# Oriented Thin Films of Insoluble Polythiophene Prepared by the Friction Transfer Technique

**DOI:** 10.3390/polym13152393

**Published:** 2021-07-21

**Authors:** Nobutaka Tanigaki, Chikayo Takechi, Shuichi Nagamatsu, Toshiko Mizokuro, Yuji Yoshida

**Affiliations:** 1AIST Kansai, National Institute of Advanced Industrial Science and Technology, Ikeda, Osaka 563-857, Japan; takechic19@chem.sci.osaka-u.ac.jp (C.T.); mizokuro-t@aist.go.jp (T.M.); 2Graduate School of Science, Osaka University, Toyonaka, Osaka 560-0043, Japan; 3Department of Physics and Information Technology, Kyushu Institute of Technology, Iizuka, Fukuoka 820-8502, Japan; nagamatsu@phys.kyutech.ac.jp; 4Research Institute for Advanced Electronics and Photonics, National Institute of Advanced Industrial Science and Technology, Tsukuba 305-8565, Ibaraki, Japan; 5Global Zero Emission Research Center, National Institute of Advanced Industrial Science and Technology, Tsukuba 305-8565, Ibaraki, Japan; yuji.yoshida@aist.go.jp

**Keywords:** polythiophene, conjugated polymers, molecular orientation, X-ray diffraction, oriented films

## Abstract

A thin film of unsubstituted polythiophene (PT), an insoluble conjugated polymer, with molecular chains uniaxially oriented in plane was prepared by the friction transfer method. The structure of highly oriented thin films of PT was investigated using grazing-incidence X-ray diffraction (GIXD), ultraviolet–visible (UV–vis) spectroscopy, and infrared (IR) spectroscopy. The polarized UV–vis and IR spectra and GIXD measurements showed the PT molecular chains were well aligned in parallel to the friction direction. The GIXD studies clarified that the polymer backbones were aligned with very narrow distribution, such that the half-width was about 4 degrees. The degree of orientation of the PT friction-transferred film was higher compared with those of regioregular poly(3-alkylthiophene)s. Moreover, the GIXD results show a preferred orientation where the *a*-axis is perpendicular to the substrate plane.

## 1. Introduction

Conjugated polymers offer excellent electrical and optical properties, reasonable chemical stability and good processability [1,2]. The arrangement of the conjugated polymer backbone in films is a very important factor for electronic and photonic device applications. The chain alignment improves the device’s performance, for example, enhanced mobility in field-effect transistors (FETs) [3]. Furthermore, it gives the devices anisotropic functions, such as polarized luminescence [4,5]. These are caused by the fact that the electrons are delocalized along the polymer backbone.

Several methods such as mechanical alignment (stretching [6] and rubbing [7], etc.), Langmuir–Blodgett deposition [8], liquid-crystalline self-organization [9], and alignment on specific substrate [10,11] have been reported for aligning conjugated polymers in one direction. One of the mechanical techniques for the arrangement of polymer backbone is the friction transfer technique, which was discovered by Makinson and Tabor in 1964 for poly(tetrafluoroethylene) (PTFE) [12]. The friction-transferred PTFE films have attracted much attention because of their ability to induce the orientation of other materials [13]. We applied this method to various polymers, especially conjugated polymers, such as polysilanes, poly(*p*-phenylene)s, poly(3-alkylthiophene)s (P3ATs), poly(*p*-phenylenevinylene)s, polyfluorenes, and polyaniline [14,15,16,17,18,19,20,21]. We also studied the use of friction-transferred polymers for fabricating some devices with anisotropic functions, e.g., anisotropic FETs [22,23], polarized electroluminescence devices [24], and polarized photodetectors [25]. The greatest merits of this method are the following two points: (1) it enables the preparation of the film for insoluble conjugated polymer; and (2) the obtained film is highly oriented.

In this study, we apply the friction transfer technique to unsubstituted polythiophene (PT). PT has environmental (air, water and high temperature) stability, structural versatility, and potentially high electrical conductivity [2,26,27]. However, similarly to most conducting polymers without any side chains, PT is insoluble in common solvents and difficult to be processed into films. The introduction of flexible side chains gives the conjugated polymers solubility and processability, which promotes research in the application of electronic devices. P3Ats in particular, which have regioregularity in the position of their side chains, show very good electrical properties for electronic devices and have become of major interest. However, few studies on unsubstituted PT for device applications have been reported [28]. So far, PT films have been fabricated during polymerization by electrochemical [28,29,30] or chemical oxidation [31]. Thermal conversion from a soluble precursor to PT films has been proposed [32]. As another method, thermal vapor deposition can offer thin films of PT, in which PT molecular chains are oriented in a normal manner to substrates [33]. However, vapor deposition has the problem that it is always accompanied with chain scission. Therefore, few detailed studies on PT films have been carried out as compared with its derivatives, namely P3AT with good solubility. Morphological and structural studies on oriented PT films are of both fundamental and applied importance [34] in understanding their own and derivatives’ physical properties in films. We successfully prepared highly oriented PT films by using the friction transfer technique. In the present paper, we will report the surface morphology and molecular arrangement in the oriented films of PT prepared by the friction transfer technique.

We have already reported usages of friction-transferred PT films as orienting substrates for other semiconducting molecules [35,36,37,38]

## 2. Materials and Methods

The PT was purchased from Aldrich Chem. Co. (St. Louis, MO, USA). The PT films were prepared on various substrates, such as fused quartz plates and silicon wafers by the friction transfer technique. A pellet was compressed from the polymer powder. The friction transfer process was carried out by squeezing and drawing the pellet of the PT on heated substrates. We searched for optimal conditions for preparation, especially temperature, which is a very important condition. Relatively good films can be obtained with substrate temperature of more than 200 °C. The typical substrate temperature was 250 °C. The temperature was higher than those of P3HT and P3DDT. The typical temperatures of P3HT and P3DDT were 150 °C and 100 °C, respectively. These polymers are soluble and have melting points. The applied load for squeezing was 20 kgf/cm^2^, and the drawing speed was 1 m/min.

Atomic force microscope (AFM) images were obtained using a SEIKO SPA-300 with a silicon cantilever of spring constant 20 N/m. Ultraviolet–visible (UV–vis) absorption spectra were measured using a Shimadzu UV-3150 spectrophotometer with a Glan–Taylor polarizing prism. Quartz was used as a substrate for UV spectroscopy. Infrared (IR) spectra were taken with a BIO-RAD FTS175C FT-IR spectrometer with a wire-grid polarizer. The resolution and number of scans were set at 4 cm^−1^ and 20,000, respectively. Silicon was used as a substrate for transmission IR spectroscopy.

The grazing-incidence X-ray diffraction (GIXD) measurements were performed at a BL13XU beamline of SPring-8 (Japan Synchrotron Radiation Research Institute (JASRI), Hyogo, Japan) equipped with ATX-GSOR (RIGAKU Co., Tokyo, Japan). The wavelength, λ, of the monochromatized incident X-ray was 0.099 nm. The beam size at incident slits was 0.1 mm × 0.1 mm square. The GIXD patterns are represented in reciprocal space with scattering vector (*Q = 4*πsin*θ/**λ*, where *θ* and *λ* are the Bragg angle and the wavelength of the incident X-ray, respectively) *Q_x_*, *Q_y_* and *Q_z_*; the scattering vector *Q_x_* is parallel to the film surface and parallel to the drawing direction as shown in Figure 1a; *Q_y_* is parallel to the film surface and orthogonal to the drawing direction; *Q_z_* is normal to the film surface. The grazing angle of incident X-ray, *ω*, is fixed at 0.14° to ensure total reflection against the substrate (*Q_xy_*). The angle, *φ*, between the scattering vector and the drawing direction of friction transfer was fixed during the in-plane *θ*–2*θ* scan; *θ* = 0° for the GIXD patterns with *Q_x_* and 90° for that with *Q_y_*. For the in-plane *θ*–2*θ* scan, the scanning speed was 2°/min and the interval step was 0.02°. The molecular orientation distribution is determined by the locking scan, which is performed at a fixed *Q* at the scanning speed of 10°/min with the interval step of 0.4°. Silicon was used as a substrate for GIXD measurements.

Furthermore, in order to evaluate the orientation of the crystallites regarding the substrate plane, energy-dispersive (ED) GIXD measurements [39,40,41] were carried out. The schematic diagram of ED-GIXD geometry is illustrated in Figure 1b. A pure germanium solid state detector (SSD) (Canberra 7935 2/S) is movable in two directions, i.e., horizontal (2*θ_H_*) and vertical direction (2*θ_V_*). With the coupling of 2*θ_H_* and 2*θ_V_*, reflections with any directions can be observed [40]. The diffraction angle 2*θ* is related to 2*θ_H_* and 2*θ_V_* as the following equation:cos2*θ* = cos2*θ*_*H*_ • cos2*θ*_*V*_

The tilt angle *ξ* is between the horizontal plane (the sample plane) and the scattering vector, which is normal to the diffracting plane. The angle *ξ* is calculated by the following equation:tan*ξ =* tan2*θ_V_*/sin2*θ_H_*

The ED-GIXD system was constructed by Rigaku Co. Ltd. The white X-rays (non-monochromatized X-rays) generated by a molybdenum (Mo) target tube (2.0 kW) were used as incident X-rays. Typically, incident X-rays were set up at a glancing angle of about 0.1°. The incident X-rays were collimated by a Soller slit with a divergent angle of 0.2°. The diffracted X-rays passed through a Soller slit with a divergent angle of 0.2° and detected by the SSD, which is an energy-dispersive detector. Moreover, by rotating the sample stage (with the sample) in the sample plane (the azimuthal rotating angle *φ*), the orientational distribution of the diffraction plane can be evaluated. The energy dispersive system is advantageous in the evaluation of crystal orientation as compared with the conventional angular dispersive one because the directions (orientation) and sizes (*d*-spacing) of the scattering vectors in the reciprocal space can be measured separately. The accumulated time of measuring an ED-GIXD spectrum was 1200 s.

## 3. Results and Discussion

### 3.1. Surface Morphology

Friction-transferred films of PT were observed with a polarized optical microscope. Under the cross Nicol condition, the whole film was seen to be bright when the friction direction was at 45° from the polarizer axes. Rotating the film changed it to dark uniformly. This suggests that the film was well-oriented uniaxially.

Each thickness of the PT film was estimated by a surface profile tracer (KLA Tencor Co., Milpitas, CA, USA). The PT film takes the thickness of less than 10 nm, which is thinner than that of poly(3-hexylthiophene) (P3HT) and poly(3-dodecylthiophene) (P3DDT), which are several tens of nm [17,41,42].

Surface morphology is observed by AFM measurements for the friction-transferred oriented films of PT. The friction-transferred film of PT shows a uniform surface morphology, as shown in Figure 2, which is remarkably different from the PTFE film surface with numerous sub-micrometer periodic grooves [11,12,13]. Obvious structure cannot be found on the PT films. The surface roughness (Ra) for the oriented film of PT is 1.38 nm, which is smooth compared to those of P3HT and P3DDT (Ra: 1.94 nm and 13.94 nm, respectively; figure not shown) and the PT films by electrochemical synthesis [30]. Friction-transferred PT films were thinner and smoother than those of P3DDT.

### 3.2. Molecular Chain Orientation

#### 3.2.1. Polarized UV–Vis Spectroscopy

Figure 3 shows the polarized UV–vis absorption spectra of the oriented film of PT that underwent the friction transfer technique for optical polarization both parallel (solid) and perpendicular (dashed) to the friction direction.

The PT film shows absorption peaks at 516, 551 and 605 nm with parallel polarization. The wavelength of these absorption bands was longer than oligothiophenes. The absorption band edge is beyond 650 nm, which is the same level as P3AT films. PTs produced by Aldrich Company have smaller molecular weights, but they are not oligomers. In general, insoluble polymers have low molecular weight. The absorption band edge of PT films shows that PT conjugation length is comparable with P3ATs and degradation of the main chain did not occur.

On the other hand, no absorption peak with perpendicular polarization was shown in the visible region, which indicates that the PT polymer backbone in the friction-transferred film is very highly oriented along the friction direction. Even the friction-transferred P3ATs showed weak perpendicular absorption, which came from the unoriented amorphous part [17]. The order parameter estimated from the dichroic ratio is close to 1 (the value of 1 means complete orientation). Such excellent uniaxial orientation of PT backbone has never been reported in the PT or P3ATs films prepared by other orientation methods, such as the rubbing technique [41,42,43,44], stretching on polymer substrate [45], and the Langmuir–Blodgett method [46]. Even the P3AT films oriented by the friction transfer technique [17] cannot attain the excellent orientation of the PT films.

#### 3.2.2. Polarized IR Spectroscopy

Polarized IR spectra were measured by polarized transmission. The polarized transmission spectra present information on the molecular orientation in the film plane.

Figure 4 shows polarized IR spectra for the C–H stretching (ν(C–H)) region. The band was split into two peaks in the transmission spectra. As a thiophene ring has two C–H bonds, two symmetries, sym and anti, exist. These peaks have dichroism that shows the molecular chain orientation. The transition moment of the symmetric band is perpendicular to the chain axis and that of the anti-symmetric band is parallel to the chain axis. Therefore, the peak at 3080 cm^−1^ is assigned to the symmetric band and that at 3060 cm^−1^ to anti-symmetric.

Figure 5a shows the C–H out-of-plane deformation band δ(C–H) in transmission and RA spectra of the PT film at room temperature [47,48]. Since the transition moment of the δ(C–H) mode is approximately normal to the thiophene ring, this band is a good probe for the direction of thiophene ring to the substrate. In the case of the P3DDT (Figure 5b), the δ(C–H) band [49] in the transmission spectra is absent, while that in RAS is obviously observed. Similar results for the P3HT friction-transferred film were obtained (figure is not shown). As RAS is sensitive for the bands with the transition moment normal to the reflection plane, the thiophene ring should be arranged parallel to the film plane. These findings confirmed layered structures of the friction-transferred P3DDT and P3HT films with the planes of thiophene ring parallel to the film surface deduced from GIXD study [17]. In contrast, for the solution-cast film of P3DDT films, the δ(C–H) band is seen in RAS and is not observed in transmission spectra (figure is not shown). This corresponds with the results of the GIXD measurements for P3HT [50]. The δ(C–H) band is good proof for the thiophene ring’s orientation in polythiophene derivatives.

On the other hand, the δ(C–H) band of PT appears in both transmission and RAS spectra, which suggests that the PT molecular arrangement is different from that of P3DDT. Furthermore, its arrangement differs from electrochemically polymerized PT film with thiophene rings perpendicular to substrates [30] and parallel to substrates [51,52]. Two possibilities are considered for the orientation. One is only uniaxial orientation without preferred orientation around the chain axis. This possibility is denied by the GIXD measurements mentioned in the next section. We suppose PT chains take the herringbone packing in the friction-transferred films as well as the molecular structure of PT reported in References [53,54,55,56]. Furthermore, since the transition moment of the δ(C–H) mode includes the perpendicular component to the polymer chain, the δ(C–H) peak with perpendicular polarization in the transmission spectra is more intense than that with parallel polarization, which corresponds with the orientation of the polymer chain along with drawing direction confirmed by UV–vis spectroscopy.

#### 3.2.3. GIXD

In order to obtain more detailed orientational information about the friction-transferred PT film, we investigated the out-of-plane and in-plane structure of the films using the GIXD.

The GIXD patterns are represented in reciprocal space *Q_x_*, *Q_y_* and *Q_z_* in Figure 6. In the out-of-plane GIXD pattern (*Q_z_*), peaks at *Q* = 14.16 Å ^−1^ (*d* = 4.44 Å) and *Q* = 16.25 Å^−1^ (*d* = 3.87 Å) are observed. These peaks are due to (110) and (200) diffractions of the orthorhombic (or monoclinic) structure with *a* = 7.80 Å, *b* = 5.55 Å and *c* = 7.98 Å, according to References [17,42,43,44,45]. As for the pattern with *Q_y_*, the *Q* = 13.92 Å^−1^ (*d* = 4.51 Å) and *Q* = 16.00 Å^−1^ (*d* = 3.92 Å) due to (110) and (200) appear. On the other hand, the pattern with *Q_x_* shows the only broad (002) reflection at *Q* = 16.05 Å^−1^ (*d* = 3.91 Å) and no peak due to (*hk*0) reflection. This confirmed that the c* axis was parallel to the *Q_x_* because the *c* axis (the molecular chain axis) was along the friction direction.

The profiles along the *Q_y_* and *Q_z_* directions are not similar to each other. This suggests that the PT crystallites should not assume simple uniaxial orientation. The orientation of crystallites contains some different orientations, but there is a preferential orientation with respect to the substrates. Here, we compare GIXD results with the X-ray diffraction profiles of powder samples [54]. In powder patterns, the (110) reflection is strongest and (200) and (210) follow it, but the (020) reflection is weak. GIXD along the *Q_z_* direction (110) seems to be weak and (200) reflection appears relatively strong. Along the *Q_y_* direction, (210) and (020) are observed to be relatively strong. These results suggest that the *a* and *b* axes have some orientation concerning the substrate plane. We have observed preferential orientations concerning the substrate in the friction-transferred films of P3ATs [40,41]. Now, we suppose PT molecules take a preferential arrangement in which the *a*-axis is normal to the substrate, the *b*-axis is in the substrate plane, and the molecular chains (*c*-axis) are in the substrate plane and aligned along the friction transfer direction, as shown in Figure 7, because of the following reasons. (1) The intensity of the (200) peak in the GIXD pattern (*Q_y_*) is remarkably weak compared to (110) peak and (2) whereas the (020) peak is not observed in ordinary X-ray profiles due to a weak intensity, this peak appears in the *Q_y_* pattern.

In order to confirm the orientation concerning the substrate plane, we performed the ED-GIXD measurements of a diffraction with changing angles between its scattering vector and the substrate plane [40]. In these measurements, the diffractions were measured not only in the ‘in-plane’ direction (*Q_x_*, *Q_y_*) and ‘out-of-plane’ direction (*Q_z_*), but some intermediate directions. In ED-GIXD, because the measurements can be performed with the fixed optics of X-rays, the orientation distribution of the crystallites concerning the substrate can be evaluated quantitatively. In this measurement system, as we used the white X-ray from a tube as the source, the scattering intensities are very weak and only the (110) reflection, which is the strongest, could be observed. Figure 7 shows a typical energy-dispersive diffraction spectrum. The scattering angle was fixed at 2*θ* = 7°. The (110) was observed at around *E* = 22 keV corresponding to *d* = 0.23 nm.

Fixing the diffraction angle 2*θ* so that the (110) reflection could be observed, GIXD spectra were measured at various tilting angles, *ξ,* that are between the scattering vectors and the substrate plane with the appropriate combination of 2*θ_H_* and 2*θ_V_*. The sample was rotated in the substrate plane so that the equatorial reflection (*hk*0) would be observed strongly. Figure 8 represents the change in (110) diffraction intensity with variation in the tilting angle *ξ*. The intensity strongly depends on the tilting angle *ξ*. These results exhibit a preferential orientation of the crystallites concerning the substrate plane. The distribution of intensity has an obvious peak around *ξ* = 25°. The distribution can be interpreted by the model mentioned before, in which the *a*-axis is normal to the substrate plane. According to this model, the (110) direction is at *ξ* = 25°. Figure 9 shows a schematic picture of the PT molecular arrangement in a crystallite of the friction-transferred film.

In order to evaluate the orientation distribution of the chain axis in the substrate plane, a locking scan of the most intense reflection 110, which is normal to the fiber axis, was performed (conventional GIXD at SPring-8). In the locking scan, we found that the diffraction patterns strongly depended on the angle *φ* between the scattering vector *Q* and the drawing direction of friction transfer. Changes in scattered intensity at a locked *Q* correspond to the distribution of orientation of the reflecting lattice plane to the drawing direction. Figure 10 shows the locking curve of the (110) diffraction plane in the *Q_y_* direction at a fixed *Q* = 13.92 Å^−1^ with a range of ±30°. The half-width of the distribution of the (110) diffraction plane was estimated to be 4°, which means that the molecular chains are excellently oriented along the friction direction in the film. This distribution of the PT films is comparable to that of PTFE films (about 3°) [40] and poly(9, 9′-dioctylfluorene) films (about 2–3°) [57].

In comparison with the friction-transferred P3AT films, the molecular distribution of the PT films is significantly narrower compared to that of the regioregular P3HT (10°), P3DDT (13°) [17], and poly(3-butylthiopnene) (P3BT) (11°) [42]. This finding is in agreement with the high molecular orientation obtained by polarized UV–vis spectra.

## 4. Conclusions

As for PT polymer, this study is the first to report such highly in-plane oriented films. The friction-transferred polythiophene (PT) films were studied in terms of surface morphology and molecular arrangement. As unsubstituted PT is an intractable polymer (insoluble and not melting), there are few studies on structure, properties and application. The friction-transferred PT films take a uniform surface morphology. The fabrication of thin films makes the unsubstituted PT useful for some electric devices. Notably, the high orientation of polymer backbone in-plane with a very narrow distribution was confirmed. These properties would provide the anisotropic properties suitable for various devices, such as field-effect transistor and photovoltaic devices, and an enhanced mobility should be expected. Moreover, friction-transferred PT films are very useful as orientation templates. PT films have the ability to induce the molecular orientation of other materials, e.g., oligothiophenes, as with friction-transferred PTFE films. The advantages of PT as orientation templates are the following. The orientation degree is higher than those of P3ATs. As PT is a conjugated polymer, the PT layers do not act as the insulation layers for device fabrication. Finally, because PT is insoluble in any solvent, solution process can be used to induce the orientation of other materials.

## Figures and Tables

**Figure 1 polymers-13-02393-f001:**
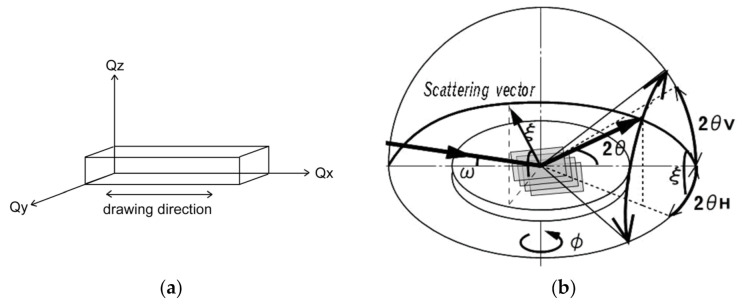
(**a**) The geometry of the scattering vectors with the drawing direction of friction transfer; (**b**) The schematic diagram of ED-GIXD geometry. The scattering angle 2*θ* is composed of horizontal angle 2*θ_H_* and vertical angle 2*θ_V_*.

**Figure 2 polymers-13-02393-f002:**
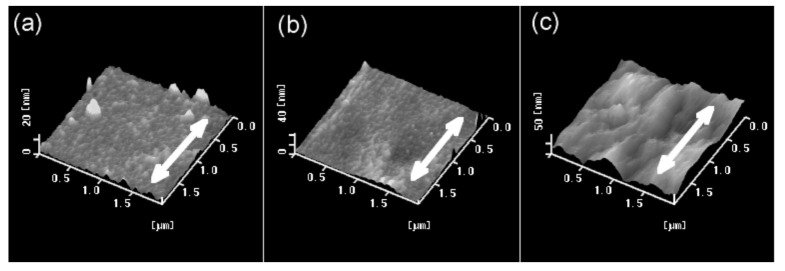
AFM images of the friction-transferred films on the glass substrates; (**a**) PT; (**b**) P3HT; (**c**) P3DDT. The drawing direction of transfer is indicated by the arrow.

**Figure 3 polymers-13-02393-f003:**
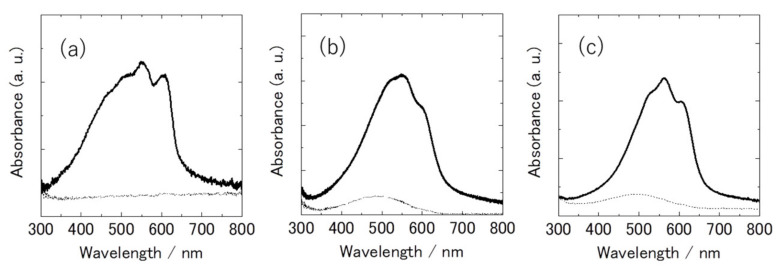
Polarized UV–vis absorption spectra of the friction-transferred PT (**a**), P3HT (**b**) and P3DDT (**c**) films. The polarization direction is parallel (solid line) and perpendicular (dashed line) to the drawing direction.

**Figure 4 polymers-13-02393-f004:**
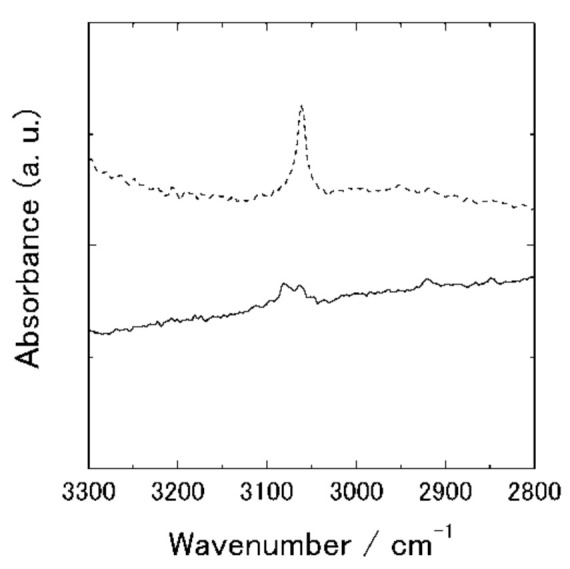
Polarized transmission IR spectra in the C–H stretching region of the friction-transferred PT film. The polarization direction in transmission spectra is parallel (solid line) and perpendicular (dashed line) to the drawing direction.

**Figure 5 polymers-13-02393-f005:**
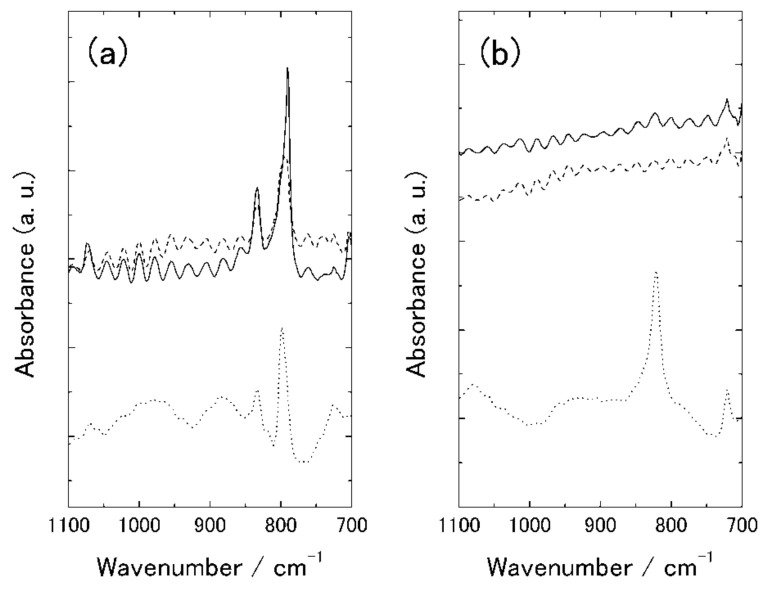
Polarized transmission IR spectra (upper) and IR-reflection absorption spectra (RAS) (lower) in the C–H deformation region of the friction-transferred PT film (**a**) and those of the friction-transferred P3DDT film (**b**). The polarization direction in transmission spectra is parallel (solid line) and perpendicular (dashed line) to the drawing direction.

**Figure 6 polymers-13-02393-f006:**
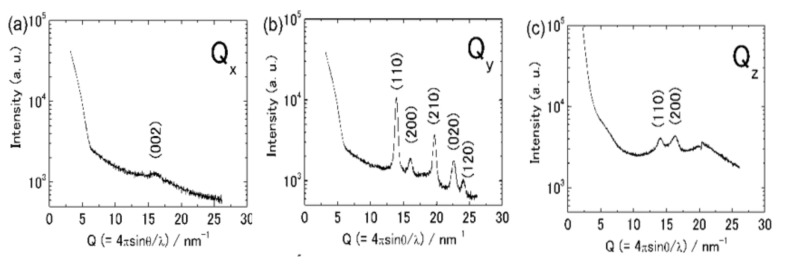
XRD profiles of friction-transferred PT film in reciprocal space *Qx* (**a**), *Qy* (**b**) and *Qz* (**c**). The direction of *Qx* is parallel to the film surface and parallel to the drawing direction. That of *Qy* is parallel to the film surface and orthogonal to the drawing direction. That of *Qz* is normal to the film surface.

**Figure 7 polymers-13-02393-f007:**
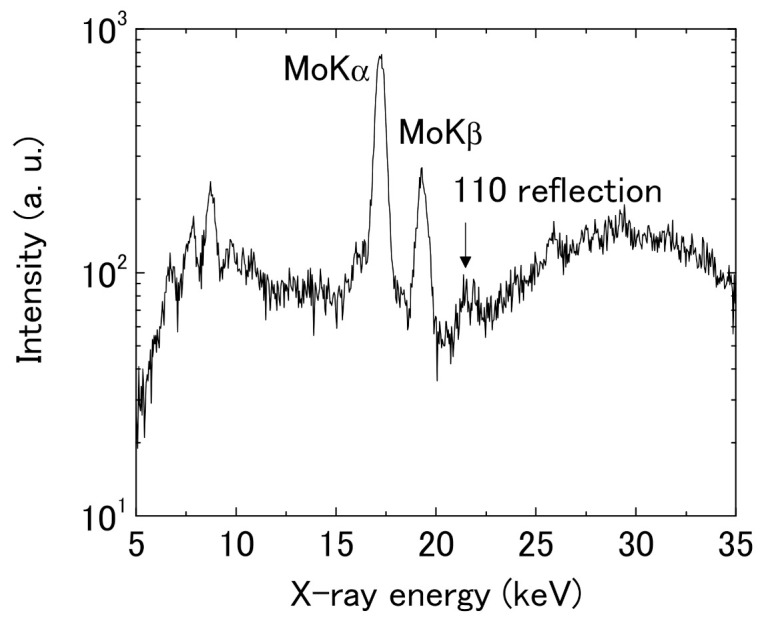
A typical energy dispersive diffraction spectrum of the friction-transferred PT film.

**Figure 8 polymers-13-02393-f008:**
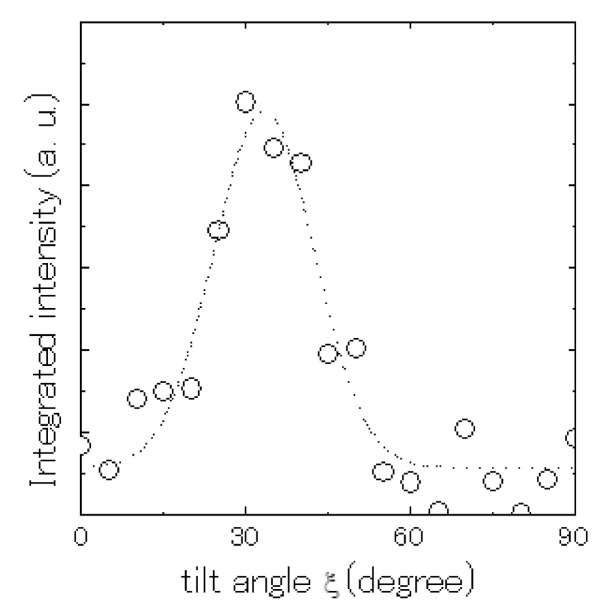
Integrated intensity of 110 reflection with variation in tilt angle *ξ*.

**Figure 9 polymers-13-02393-f009:**
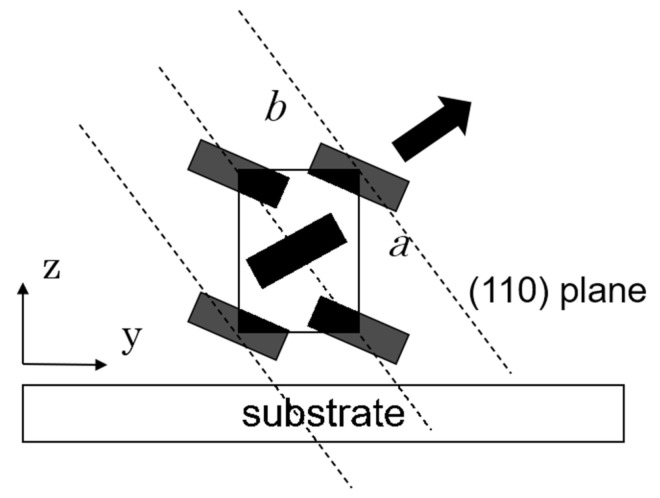
Schematic picture of PT molecular arrangement in the friction-transferred film. The friction direction, which is coincided with the chain axis of PT, is normal to the paper plane. Rod-shaped symbols are for projected PT molecules seen from the chain axis.

**Figure 10 polymers-13-02393-f010:**
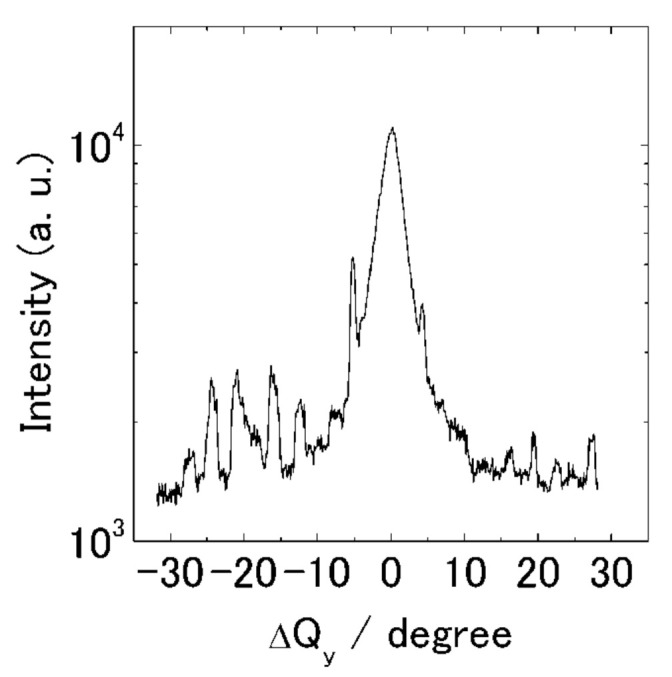
GIXD locking curve of (110) diffraction at a constant *Q* = 13.92 Å^−1^ in the *Q_y_* direction.

## Data Availability

Data is contained within the article.

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
