# Peer review of "Oriented Thin Films of Insoluble Polythiophene Prepared by the Friction Transfer Technique"

_polymers, 2021, doi:10.3390/polym13152393_

Round 1
Reviewer 1 Report
the manuscript is interesting but the authors only describe their results, in their writing there is no comparison or relevance of the results of other authors, there is no discussion of the results therefore the article lacks a strong conclusion
Images should be arranged in line with the text and increase its resolution
Author Response
Thank you very much for the valuable comments of the referees to our paper. I am returning herewith the above manuscript revised.
Comparison with P3HT and P3DDT is added in Section 2 materials and methods, 3.2.1 polarized UV vis spectroscopy, and 4 conclusion.

Reviewer 2 Report
Tanigaki provide a manuscript outlining their studies of frictional ordering of polythiophene films. The work is straightforward, reasonably well written and the conclusions are justified by the data presented. The result, that the frictional process leads to film orientation, is not surprising in and of itself, however the degree of orientation seems fairly high compared to other comparable materials. I have little to offer in the way of technical comments, as I think these are well explained and common tests. I think the work should be published, though I suggest the authors consider a few comments below before finalizing the manuscript.
First, this study seems like it is really a comparison between PT, P3HT and P3DDT, rather than just PT. Maybe this should be described more clearly in the abstract and introduction. Likewise, a few of the latter figures could be more complete with additional data from P3DDT.
Next, with respect to the UV vis absorption curves - What are the absorption peaks for the PT film related to? What is the molecular weight (as you mention here it is short) and how does it compare to the molecular weight of the other two (comparison) polymers? Obviously, long chains will have more amorphous phase in almost any crystallization scenario.
What is the thermal breakdown like for this polymer? The transfer process is relatively high temperature, does it lead to any degradation of the chains?
I list a few minor things below:
Line 79 page 2 “which is very important condition.” Should be “which is a very important condition”.
Somehow figure 1 overlaps the text.
Figure 2 shows afm scans of 3 different types of oriented polymer films, in order to show the smoothness of the PT film. However, the height scales are different in each image, which makes comparison difficult. The films should be rescanned or replotted with the same height scale to make comparison easy. As it stands, it looks like the PT film is rougher than the P3HT film.
Figure3. It is very hard to tell the ‘dashed’ curve from the solid curve. It is conceptually clear which is which, but maybe a larger linewidth on the solid curve would help differentiate.
Line 209 page 6 “Contrastively” I am pretty sure is not an English word.
Figure 9, you should label the z axis as you show it in the diagram.
The instructions preceding the references should likely be removed before publication.
Author Response
Thank you very much for the valuable comments of the referees to our paper. I am returning herewith the above manuscript revised, and the following are the answers to the comments.
First, this study seems like it is really a comparison between PT, P3HT and P3DDT, rather than just PT. Maybe this should be described more clearly in the abstract and introduction. Likewise, a few of the latter figures could be more complete with additional data from P3DDT.
- This study mainly mentioned about PT. Friction transfer films of P3HT and P3DDT were already published Ref. 17, 22 and 23.
We add some discussion of comparing with P3HT and P3DDT in Section 2 materials and methods, 3.2.1 polarized UV vis spectroscopy, and 4 conclusion
Next, with respect to the UV vis absorption curves - What are the absorption peaks for the PT film related to? What is the molecular weight (as you mention here it is short) and how does it compare to the molecular weight of the other two (comparison) polymers? Obviously, long chains will have more amorphous phase in almost any crystallization scenario.
- UV-vis absorption is assigned to the π-π* transition. Unfortunately, we do not have molecular weight information of PT, P3HT and PDDT.
What is the thermal breakdown like for this polymer? The transfer process is relatively high temperature, does it lead to any degradation of the chains?
- We consider that no thermal degradation occurred because UV absorption showed maintaining the conjugation length.
Line 79 page 2 “which is very important condition.” Should be “which is a very important condition”.
- corrected
Somehow figure 1 overlaps the text.
- corrected
Figure 2 shows afm scans of 3 different types of oriented polymer films, in order to show the smoothness of the PT film. However, the height scales are different in each image, which makes comparison difficult. The films should be rescanned or replotted with the same height scale to make comparison easy. As it stands, it looks like the PT film is rougher than the P3HT film.
- Unfortunately, samples, AFM, and original date were lost. AFM image cannot be improved and cannot be re-scanned. The numerical roughness of P3HT is added. Line151 smoother than those of P3Ats => P3DDTs.
Figure3. It is very hard to tell the ‘dashed’ curve from the solid curve. It is conceptually clear which is which, but maybe a larger linewidth on the solid curve would help differentiate.
- Figure 3 was improved.
Line 209 page 6 “Contrastively” I am pretty sure is not an English word.
- Contrastively => In contrast
Figure 9, you should label the z axis as you show it in the diagram.
- corrected
The instructions preceding the references should likely be removed before publication.
- removed
